# NO and Heme Proteins: Cross-Talk between Heme and Cysteine Residues

**DOI:** 10.3390/antiox12020321

**Published:** 2023-01-30

**Authors:** Cinzia Verde, Daniela Giordano, Stefano Bruno

**Affiliations:** 1Institute of Biosciences and BioResources (IBBR), National Research Council (CNR), Via Pietro Castellino 111, 80131 Napoli, Italy; 2Department of Marine Biotechnology, Stazione Zoologica Anton Dohrn (SZN), Villa Comunale, 80121 Napoli, Italy; 3Department of Food and Drug, University of Parma, 43124 Parma, Italy; 4Biopharmanet-TEC, University of Parma, 43124 Parma, Italy

**Keywords:** S-nitrosylation, nitrosation, nitric oxide, heme proteins, heme, cysteine

## Abstract

Heme proteins are a diverse group that includes several unrelated families. Their biological function is mainly associated with the reactivity of the heme group, which—among several other reactions—can bind to and react with nitric oxide (NO) and other nitrogen compounds for their production, scavenging, and transport. The S-nitrosylation of cysteine residues, which also results from the reaction with NO and other nitrogen compounds, is a post-translational modification regulating protein activity, with direct effects on a variety of signaling pathways. Heme proteins are unique in exhibiting this dual reactivity toward NO, with reported examples of cross-reactivity between the heme and cysteine residues within the same protein. In this work, we review the literature on this interplay, with particular emphasis on heme proteins in which heme-dependent nitrosylation has been reported and those for which both heme nitrosylation and S-nitrosylation have been associated with biological functions.

## 1. NO and Heme Proteins

Heme proteins (or hemeproteins or hemoproteins) are a structurally and functionally diverse group of metalloproteins exhibiting the heme moiety—an iron-coordinated porphyrin ring—as the prosthetic group [1,2]. The most abundant heme types, differing in the substituents of the porphyrin ring, are heme *b* (or protoheme IX), found in globins, peroxidases, and cyclooxygenases; heme *c*, typical of cytochrome c; heme *a*, found in cytochrome c oxidase; and the bacterial heme *o* [1] (Figure 1). All hemes contain an iron ion coordinated to the porphyrin pyrrol rings (Figure 1). The iron is typically Fe(II) (ferrous) or Fe(III) (ferric), with Fe(IV) (ferryl) being involved in some catalytic intermediates. One or two side chains of the protein can bind the iron ion as axial ligands (Figure 1). The heme is said to be pentacoordinated and hexacoordinated, respectively. Heme proteins encompass a wide variety of folds [2], ranging from all α-helical folds, such as globins [3,4], to all β-barrel and mixed α-helical-β-barrel structures, such as nitrophorins (NPs), α1-microglobulin, and nitrobindins [5]. Despite the large variability in their structure, the function of heme proteins is associated with one of the many intrinsic reactivities of the heme group, ultimately linked with its ‘electron sink’ nature, which makes it capable of delocalizing radical electrons. This general reactivity can lead to different functions, including electron transport, catalysis, or the reversible binding of the gaseous ligands, nitric oxide (or nitrogen oxide or nitrogen monoxide, NO), oxygen (O_2_), and carbon monoxide (CO).

Evolutionary unrelated heme proteins can play a similar physiological role, as in the case of cytochrome *a*, cytochrome *b*, and cytochrome *c*—all involved in electron transport—or nitrobindins and NPs—both involved in NO transport [6]. Conversely, heme proteins belonging to the same family can be associated with different functions, as in the case of globins, whose biological roles range from NO scavenging to O_2_ transport [7,8]. Therefore, the protein fold does not necessarily define the reactivity and biological function of heme proteins. It is the protein structure, the axial ligands that coordinate the heme iron, the redox potential, the heme accessibility from the bulk solution, and the environment around the heme that dictate the functional properties of the specific protein under physiological conditions and, hence, its biological role [2,9]. This ‘promiscuity’ can sometimes hinder the identification of the relevant physiological role in newly discovered proteins. Examples are neuroglobin (Ngb) and cytoglobin (Cygb), identified more than 20 years ago but still with no definitive indication of their prevalent function [10].

Among the typical reactivities of heme proteins is the binding to NO, followed by either dissociation or chemical modification [11] (Figure 2a). NO is a free radical that reacts with biological molecules with unpaired orbital electrons, including transition metal ions such as the heme iron [11]. The interest in these reactions has greatly increased since NO was identified as the previously elusive endothelium-derived relaxing factor (EDRF), a vertebrate biological messenger that plays a role in various biological processes [12]. The discovery that NO, CO, and H_2_S are essential in signaling pathways is a relatively recent development [13]. The biology of NO should be considered in parallel with H_2_S and CO since these small gaseous molecules exert similar signaling or messenger roles, with increasing evidence of a cross-talk between them [13].

NO is biosynthesized endogenously by heme proteins named NO synthases (NOSs), which oxidize the guanidino group of L-arginine into L-citrulline and NO in the presence of O_2_ and reduced nicotinamide-adenine-dinucleotide phosphate (NADPH) [14]. It is now accepted that NO can also be produced in vivo from nitrite (NO_2_^−^) by heme proteins exhibiting the nitrite reductase activity (NiR) [15] (see below), which has been identified in vitro for at least ten human heme proteins [16]. Therefore, heme proteins are central in NO biosynthesis.

Once generated, NO can freely diffuse across biological membranes far from the site of its generation [17,18]. Overall, the half-life of NO ranges from milliseconds to seconds depending on the tissue and its oxygenation [19]. It is limited by the interaction of NO with surrounding radicals, biomolecules—particularly heme proteins (see below) —and with O_2_ (Reaction 1). However, the latter reaction has an estimated rate constant of around 10^6^ M^−2^ s^−1^ and at physiological concentrations of NO (10 nM–1 µM) is very slow (9–900 min) [20].

2 NO + O_2_ → 2 NO_2_
(Reaction 1)


NO can have different biological fates, usually terminating with the interaction with proteins involved in intracellular signaling, triggering the activation of a wide variety of pathways. These interactions result in intracellular effects ranging from neurotransmission to immune responses [21]. Soluble guanylate cyclase (sGC), also a heme protein, is considered the primary target of NO.

In addition to heme binding, NO and its derivatives can react with amino acid side chains, a reactivity not shared by the other heme ligands, CO and O_2_ (Figure 2b). Particularly, cysteine (Cys) nitrosation (or S-nitrosylation, as the reaction is now more commonly known) to nitroso Cys (Cys-SNO) is a post-translational modification that regulates cellular mechanisms ranging from enzyme activity to apoptosis, cellular metabolism, membrane trafficking, protein phosphorylation, transcription-factor activity, receptors, and ion-channels activity [22]. This reaction is chemically different from the binding of NO to the heme group, since it requires the transfer of an electron to an acceptor. Indeed, S-nitrosylation is formally the reaction of the nitrosonium ion (NO^+^) with Cys thiolate (-S^−^) or the reaction of the Cys thiyl radical (-S^·^) with NO, although the latter mechanism is unlikely since the thiyl radical rapidly reacts with O_2_ [23]. However, NO can be oxidized non-enzymatically to Cys-reactive derivatives, allowing S-nitrosylation.

In this review, we will focus on the dual reactivity of heme proteins with NO. These two reactivities are worth investigating jointly for heme proteins for several reasons:

(i) Metal-to-Cys S-autonitrosylation involves both the heme group and a specific Cys residue of the same protein, making concerted nitrosylation and S-nitrosylation a complex mechanism for post-translational modifications.

(ii) Allosteric communication between the heme moiety and the reactive Cys residue has been observed, making NO-based chemistry a dual source of conformational and functional complexity for heme proteins.

(iii) NO, a signaling molecule with several roles, can be freed by heme proteins either through dissociation from the heme group or through S-transnitrosylation from Cys-SNO. The two effects are difficult to investigate separately, especially in vivo, where relative concentrations of NO and other reactive nitrogen compounds are challenging to measure. For instance, there is an ongoing debate regarding the molecular source of NO in red blood cells (RBCs), which is responsible for vasodilation under hypoxic conditions. It was proposed to either derive from hemoglobin (Hb) Cysβ93-SNO or through the reduction in NO_2_^−^ at the Hb heme (see below).

(iv) In heme proteins, the NO derivatives freed by Cys-SNO encounter a heme group in their proximity—and vice versa—raising the possibility of immediate scavenging and poor diffusion outside of the protein matrix. When assessing NO delivery by heme proteins, the reactivity of both the heme group and the Cys residues need to be characterized both in vitro and in vivo.

(v) Experimentally, NO and its derivatives are relatively unstable and can undergo several chemical and biochemical reactions, thus making it challenging to dissect the reactivity of the heme group and Cys residues within the same protein. Indeed, some ambiguity about the specific reaction between the protein and NO is present in the less recent literature, even in in vitro studies. The physiological occurrence and relevance of either reaction at the heme group and Cys residues still need to be assessed for each heme protein.

## 2. General Reactivity of Heme Proteins with NO

### 2.1. Reactivity of NO with the Heme Group

Recent reviews on heme-binding proteins extensively cover the complex reactivity of NO with the metal center of the heme group [24,25]. For several heme proteins, the interaction or the reactivity with NO is widely accepted as physiologically relevant, whereas for others—although intrinsically reactive with NO in vitro—the relevance is under debate. For circulating proteins such as Hb, several reactions might be relevant in vivo at different times, depending on the local microenvironment they encounter within the bloodstream. The main reactivities that heme proteins exhibit are reported below and in Figure 3a.

*Reversible binding to ferrous heme.* Under anaerobic conditions, Fe(II) heme proteins rapidly bind NO to produce a stable nitrosyl complex (Reaction 2, R2 in Figure 3a). For instance, Hb exhibits a kNO of 60 µM^−1^s^−1^ [26]. The binding is reversible, but the dissociation rate constants are typically very small, with koff < 10-3 s^−1^ [27,28].

P-Fe(II) + NO → P-Fe(II)-NO
(Reaction 2)


In vivo, this reaction is associated with NO scavenging only under anaerobic conditions, since the presence of heme-bound O_2_ would quickly result in NO dioxygenation (see Reaction 4).

*Reversible binding to ferric heme.* The Fe(III) heme group forms bonds 6–7 orders of magnitude weaker with NO in comparison with Fe(II) (Reaction 3, R3 in Figure 3a) [24,25]. For Mb, the k_NO_ for this reaction is 0.070 µM^−1^s^−1^ [29,30], with k_off_ typically >10 s^−1^.
P-Fe(III) + NO → P-Fe(III)-NO(Reaction 3)

Some heme proteins play a physiological role based on this reaction. For instance, NPs are heme-binding lipocalins found in the saliva of the bloodsucking Reduviidae and Cimicidae families of Heteroptera. They are stable in the Fe(III) state, and the structure of the distal side of the heme group stabilizes the NO complex and prevents reactions with O_2_, water, hydroxides, or thiols, thus allowing their function as transporters of NO [31]. NO is then released at the feeding site, where it induces vasodilation.

*NO dioxygenation.* In the presence of O_2_, several Fe(II) heme proteins, and particularly globins, are known to catalyze the NO-dioxygenase (NOD) reaction, where NO is oxidized to nitrate (NO_3_^−^) via a peroxynitrite intermediate (Reaction 4, R4 in Figure 3a):P-Fe(II)- O_2_ + NO → P-Fe(III) N(O)OO → P-Fe(III) + NO_3_^−^(Reaction 4)

This reaction, which results in heme oxidation, is very fast (Hb k_NOD_ 60–80 µM^−1^s^−1^ [26]) when the heme group is bound to O_2_ and exposed to NO. When the heme protein is bound to NO and exposed to O_2_, the reaction occurs at a much slower rate, since NO has to dissociate to allow O_2_ binding before the reaction can take place. The NOD reaction in vivo is typically associated with NO scavenging under aerobic conditions [26], but also, as in the case of *Ascaris* Hb, to O_2_ scavenging under anaerobic conditions [32]. It has been noted that catalytic reactions that lead to heme oxidation can be physiologically relevant only if efficient mechanisms for its reduction are in place to restore the Fe(II) form [33].

*Autoreduction in Fe(III)–NO complexes.* Fe(III) heme proteins undergo ‘reductive nitrosylation’ (or ‘autoreduction’) in the presence of NO (Reaction 5, R5 in Figure 3a).
P-Fe(III) + NO + OH^−^ → P-Fe(II)-NO_2_H → P-Fe(II) + NO_2_^−^ + H^+^(Reaction 5)

The reaction occurs in two steps and is pH-dependent, since it usually requires the displacement of the intermediate by hydroxide [34] (Figure 1). Since the affinity of heme proteins in the Fe(III) form for NO is usually low (see Reaction 3), this reaction requires NO concentrations that afford a significant binding. 

*Nitrite reduction.* The reverse of Reaction 5, i.e., the reduction in NO_2_^−^ (NiR) by Fe(II) heme in the absence of O_2_ (Reaction 6, R6 in Figure 3a), has been extensively investigated in mammals because of its potential to produce NO [35].
P-Fe(II) + NO_2_^−^ + H^+^→ P-Fe(III) + NO + OH^−^(Reaction 6)

At least ten mammalian heme proteins have been characterized in vitro for NiR, i.e., Hb, myoglobin (Mb), Ngb, Cygb, cytochrome c oxidase, cytochrome bc1, cytochrome c, endothelial (eNOS), cytochrome P450, and indoleamine 2,3-dioxygenase 1 [8,16,36,37]. For proteins that are physiologically in the Fe(II) state and exhibit high concentrations (i.e., Mb and Hb in muscle tissue and red blood cells, respectively), the extent of NO production through this reaction is under debate, since newly produced NO would tightly bind the nearby Fe(II) hemes (Reaction 1).

NO_2_^−^ concentration in mammalian tissues is fairly high (0.1–10 μM [38]), confirming that the NiR reaction can physiologically take place in vivo. NO_2_^−^ can have a dietary origin [39] or can be produced from nitrate (NO_3_^−^) by bacterial Mo-containing nitrate reductases [40], suggesting a NO_3_^−^-NO_2_^−^-NO pathway for NO biosynthesis that involves the oral microbiome and enteral symbiotic bacteria for the initial step [41,42].

### 2.2. S-Nitrosylation

S-nitrosylation is a Cys post-translational modification that reversibly regulates protein function and is a fundamental regulatory mechanism in NO-related signaling [22,43]. It consists of the covalent binding of a nitroso group to a Cys residue to form the SNO. S-nitrosylation may produce novel protein–protein interactions by producing changes in the surface-charge or by allosterically generating solvent-exposed binding sites [44]. The current view is that S-nitrosylation plays a role equivalent to phosphorylation or acetylation to elicit the biological responses of target proteins, and the modification is often considered as an evolutionarily conserved signaling mechanism that involves many classes of proteins located in the cardiovascular system [45].

Regardless of the reaction responsible for S-nitrosylation, it is believed that not all exposed Cys residues can form stable SNOs. The identification of S-nitrosylation sites in the proteome [46] has allowed the identification of Cys environments that either promote the formation of SNOs or stabilize them once they are produced. Particularly, the presence of an acid-base motif in the proximity of the reactive Cys was associated with the formation of a thiolate, thus favoring the reaction [47]. Since the environment of Cys side chains can depend on the conformation of the protein within its conformational space, the residue can be reactive in one conformation but not in another. For example, it was proposed that in oxygenated Hb (R-state), the histidine (His) residue proximal to Cysβ93 favors the base-catalyzed S-nitrosylation, whereas, in the deoxygenated form (T-state), denitrosylation is facilitated by the proximal aspartic acid (Asp) residue [47]. In serine racemase, the reaction with either NO or nitroso donors is also conformation-dependent and occurs only in the conformation stabilized by its allosteric effector ATP [48]. Hydrophobic environments, such as biological membranes or environments formed by protein structures, favor the formation of SNOs ([47] and references herein). 

In the last two decades, S-nitrosylation has drawn attention mostly for the biomedical implications for its direct effect on a variety of signaling pathways in physiological and pathological conditions [49]. In several neurodegenerative diseases, high levels of NO may contribute to mitochondrial dysfunction and protein misfolding via aberrant protein S-nitrosylation. Aberrant protein S-nitrosylation may particularly impact the quality control of protein folding, including molecular chaperones, autophagy, and the ubiquitin-proteasome system [49].

The main reactions in which Cys residues and NO are involved have been summarized in Figure 3b and detailed below.

*S-nitrosylation associated with reduction.* It is accepted that the direct reaction of the NO radical with thiols does not result in S-nitrosylation since the reaction requires the loss of an electron. However, in the presence of O_2_ or other oxidants, a redox reaction can occur, with the formation of the SNO through a thiyl radical or through a nitrosonium (NO^+^) ion [22] (Reaction 7, R7 in Figure 3b).
P-SH + NO → P-SNO + e^−^ + H^+^(Reaction 7)

*S-nitrosylation by transnitrosylation.* S-nitrosylation can also result from transnitrosylation by S-nitrosoglutathione (GSNO) [43] or S-nitrosylated proteins [50,51] (Reaction 8, R8 in Figure 3b).
P-S-NO+ P′-SH → P-SH + P′SNO(Reaction 8)

Transnitrosylation has been associated with site-specific protein–protein interactions mediated by specific recognition motifs [47] (and references herein) (Figure 2). Enzymes that donate and accept nitroso groups from proteins are named S-nitrosylases and denitrosylases, respectively, and can have high-molecular weight final acceptors—as the thioredoxin/thioredoxin reductase system (Trx1/TrxR) —or low molecular weight acceptors, as the GSNO reductase and SNO-coenzyme A reductase systems [52,53]. The enzymatic control of S-nitrosylation is fundamental for the regulation of NO-based signal transduction and is dynamically regulated by the equilibrium between *S*-nitrosylated proteins and low-molecular-weight SNOs, which may be a cause or consequence of diseases, including asthma, cystic fibrosis, Parkinson’s disease, heart failure, and stroke [47].

*S-nitrosylation by reaction with nitrous anhydride.* S-nitrosylated Cys residues can also be formed by the reaction with dinitrogen trioxide (or nitrous anhydride, N_2_O_3_) (Reaction 9, R9 in Figure 3b). N_2_O_3_ partially dissociates, forming the nitrosonium ion (NO^+^), which reacts with thiolates (Reaction 8, transnitrosylation). N_2_O_3_ is formed through the reaction of NO with O_2_, particularly in hydrophobic microenvironments such as cell membranes [54].
P-S^−^ + N_2_O_3_ → P-SNO+ NO_2_^−^(Reaction 9)

*Metal-to-Cys S-nitrosylation.* Metal-to-Cys S-nitrosylation (or heme-dependent S-nitrosylation or heme-assisted S-nitrosylation in the specific case of heme proteins) is a special case of S-nitrosylation associated with reduction (Reaction 7) and is of great relevance to heme proteins. It consists of the transfer of a nitroso group from a heme group to a nearby Cys residue, either within the same protein (auto S-nitrosylation) or in another protein or peptide (Reaction 10, R10 in Figure 3).
P-Fe(III)-NO+ P-SH → P-SNO + P-Fe(II) + H^+^(Reaction 10)

The NO^+^ ion is generated by the binding of NO to a Fe(III) heme (Reaction 3), which is reduced to Fe(II) (Figure 3).

## 3. Heme Reactivity and S-Nitrosylation of Heme Proteins

In this review, we will describe the reactivity of NO at the heme site and at Cys residues for heme proteins for which both have been described. An overview is in Table 1. In case they are available, the structures of S-nitrosylated proteins are depicted in Figure 3.

### 3.1. Hemoglobin

Hb is an O_2_ transporter contained in RBCs at a high concentration. It is a heterotetramer of paralogous subunits exhibiting the globin fold [103]. The subunits of the most abundant human form, HbA, are named α and β and each binds a Fe(II) heme coordinated to a His residue named proximal His (F8 in the globin nomenclature). The residues that coordinate and orient the heme group are among the few invariant residues among all Hbs [103]. Hb exhibits cooperative binding associated with the equilibrium between at least to conformations endowed with a different affinity for O_2_. The binding of O_2_ at one subunit shifts the proximal His with respect to the heme, triggering a modification of the network of non-covalent bonds at the α_1_β_2_ interface. These changes destabilize the conformation known as the “T-state”, endowed with low O_2_ affinity and stabilize the conformation known as the “R-state”, endowed with high O_2_ affinity [104].

#### 3.1.1. Heme Reactivity with NO

Among heme proteins, the NO adducts of Hb are some of the most studied systems in biology. Hb in the Fe(II) state binds NO (Reaction 2, binding of NO to Fe(II)) with a much higher affinity in comparison with O_2_ and CO, particularly because of a slow dissociation rate, in the range 10^−5^–10^−3^ s^−1^ [55]. The binding of NO has been characterized in detail [56], indicating a cooperative process and estimating different dissociation rates for the two different subunits and the two quaternary states [56]. Although widely investigated in vitro [57], the formation of significant amounts of the Hb–NO complex in vivo is unlikely, given the high concentrations of the competing ligand, O_2_, to form Hb-O_2_.

Hb-Fe(II)-O_2_ quickly reacts with NO through the NOD reaction (Reaction 4) to yield Hb-Fe(III) and nitrate. This reaction has been associated with the scavenging of NO and vasoconstriction, as observed for hemolytic diseases [8,16,36,37,61]. Under physiological conditions, this reaction is limited by the confinement of Hb inside RBCs.

The autoreduction in Fe(III)-NO (Reaction 5) was observed for Hb in 1901 by Haldane, who prepared Hb-Fe(II)-NO by adding NO_2_^−^ to Hb [62]. This activity, which was first characterized in cured meat [62], was later suggested to play an important role in delivering NO to tissues in the capillaries, where Hb—mostly in the deoxy form—can produce NO through the NiR (Reaction 6) [63].

Impaired reactivity of Hb with NO has been associated with pathology and was particularly studied in the field of Hb-based O_2_ carriers (HBOCs), a class of biotechnological therapeutics based on chemically or genetically modified heme proteins designed to deliver O_2_ to tissues as an alternative to RBCs [61,105]. One of the hurdles in this approach consists of the scavenging of NO by Hb derivatives when administered intravenously in a cell-free form. Indeed, HBOCs are not shielded by RBCs, which greatly reduce the interaction of Hb with NO [61]. This interaction—which can either occur through the binding of NO to Fe(II) (Reaction 2) or through the autoreduction in Fe(III)–NO complex (Reaction 5) depending on the saturation of Hb—interferes with the regulation of the vascular tone by NO signaling, typically causing life-threatening vasoconstriction [26]. NO scavenging has been addressed by producing sufficiently large polymeric Hbs to reduce extravasation and consequent reactions with NO or by creating recombinant Hbs with reduced rates for the NOD reaction (Reaction 4).

#### 3.1.2. S-Nitrosylation

HbA is by far the better-characterized globin for which S-nitrosylation has been detected, with an extensive functional and structural characterization [106]. It was not only one of the first proteins for which S-nitrosylation was identified, but also the first one for which a physiologically relevant role for S-nitrosylation was proposed [107]. Indeed, it was hypothesized that Hb is S-nitrosylated in the lungs and releases the nitroso group in the capillaries, producing vasodilation and regulating blood pressure under hypoxic conditions, an effect long known but of unclear molecular origin [108].

All mammalian Hbs contain a reactive Cys at position β93, the primary site of Hb S-nitrosylation. Cysβ93 is solvent-accessible, close to the heme group, and is a known target for thiol reagents, which can allosterically modulate O_2_ affinity, cooperativity, and Hb sensitivity to pH [109,110]. Cysβ93 was implicated in several redox reactions (recently reviewed in [111]) and it was particularly proposed as a scavenger of reactive oxygen species (ROS). Consistently, mice inflammation models expressing the Cysβ93Ala variant of Hb showed a greater degree of hypotension and lung injury associated with the increased formation of reactive species in RBCs [112].

Hb S-nitrosylation of Cysβ93 was suggested to be possible through a heme-dependent mechanism in which either Fe(III)-NO heme mediates the metal-to-Cys S-nitrosylation of Cysβ93 (Reaction 10) or NO_2_^−^ reacts with Fe(II) heme through the NiR (Reaction 6) to produce NO, which in turn can react with the –SH radical (Reaction 7) [58] (Figure 4a). This reaction was suggested to be conformation-dependent, since it was reported to take place at a higher rate in the lungs, where Hb is mostly in the R-state [59]. Indeed, the total NO bound to Hb was shown to be constant irrespective of pO_2_ during the arterial–venous cycling, whereas the relative amount of NO bound to hemes and Cysβ93 changes. It was concluded that NO is exchanged between hemes and thiols [59]. According to this model, upon the conformational transition to the T-state in the capillaries, NO is transnitrosylated to other target proteins or peptides [108] (Figure 4). The physiological relevance of this reaction is under debate since it would require Hb in the Fe(III) state, usually maintained at low concentrations by the antioxidant systems known to operate within RBCs [113]. The His residue proximal to the Cys93 in the β subunit of Hb in the R-state was suggested to facilitate S-nitrosylation, whereas the proximal aspartic acid (Asp) in the deoxygenated T-state would favor denitrosylation [114]. Based on these observations, Stamler and colleagues proposed that SNO-Hb in erythrocytes serves as a physiological regulator of vascular tone by preserving the bioactivity of NO within the vasculature to modulate the blood flow [59].

Metal-to-Cys S-nitrosylation was shown to be absent in the Hb pathologenetic variant —HbS—responsible for sickle cell anemia, suggesting that defective NO processing might contribute to the disease [115]. Based on this model, a mixture of HBOCs with their S-nitrosylated derivative has been investigated, concluding that they might be useful in the treatment of conditions of acute hypoxia, such as tumor oxygenation, wound healing, hemorrhagic trauma, and sickle cell and hemolytic anemia [116].

Following the characterization of mice expressing the Cysβ93Ala Hb variant, the role of Cysβ93 S-nitrosylation as the mechanism responsible for NO-mediated vasodilation has been put into question. Some NO-mediated effects such as ex vivo cardioprotection, inhibition of platelet activation, or capillary relaxation were demonstrated to be intact in these mice [117,118], although hypoxic vasodilation was critically impaired. Altogether, these studies supported the hypothesis that NO bioactivity occurs independently of S-nitrosylation and that Reaction 6 (NiR reaction) better explains RBC-associated NO vasoactivity [117]. Particularly, it has been shown in several experimental settings that NO_2_^−^ mediates the inhibition of platelet activation in the presence of RBCs and that NO scavengers abolish this effect [119].

The extent of heme-assisted S-nitrosylation in vivo has also been challenged by showing that Hb-SNO—although measurable in the human circulation—did not exhibit a significant arterial–venous gradient and that NO delivery from Hb-SNO was small [63]. Heme-assisted S-nitrosylation has also been disputed [60]. Indeed, it was shown that Hb is not capable of transferring NO from Cysβ93 to the heme—and vice versa—upon T–R conformational changes, neither in whole blood, nor in washed red cells or purified Hb [60].

The conflicting reports and models on NO release or production from RBCs [120] confirm the difficulty in dissecting the reactivity of the heme group of heme proteins with respect to the S-nitrosylation of Cys residues, calling for a thorough functional and structural characterization of heme proteins for which these reactivities are proposed as physiologically relevant.

### 3.2. Ascaris Hemoglobin

*Ascaris lumbricodes* is a parasitic nematode that, in the adult form, inhabits the O_2_-poor small intestine, leading an anaerobic life. It currently infects nearly 25% of the world’s human population [121]. The octameric *Ascaris* Hb, unlike tetrameric vertebrate Hbs, is free in solution in the perienteric fluid of the worm and it is endowed with an exceptionally high affinity for O_2_, 25,000 times higher than human Hb. The globin domain of *Ascaris* displays the classic globin-fold with minor differences [64]. The distal His residue, characteristic of many vertebrate Hbs, is replaced by a glutamine residue, which dramatically increases O_2_ affinity.

#### 3.2.1. Heme Reactivity with NO

*Ascaris* Hb works as a dioxygenase by catalyzing NO_3_^−^ production from O_2_ and NO through the NOD reaction (Reaction 4), thus maintaining the perienteric fluid hypoxic [65]. This reaction is uncommon since vertebrate globins usually scavenge NO through Reaction 4 in the presence of an excess of O_2_. In the O_2_-poor environment where *Ascaris* lives, the opposite occurs, with NO being produced in excess by NOS to detoxify O_2_ [32]. Upon oxidation, *Ascaris* Hb-Fe(III) subsequently binds NO through Reaction 3 [65].

#### 3.2.2. S-Nitrosylation

Cys72 (E15 in globin nomenclature) is positioned in the distal pocket, close to the heme [122] and is found mostly as nitrosothiol in the perienteric fluid of *Ascaris*. Unlike other globins, nitrosothiols oxidize the heme group of *Ascaris* Hb, suggesting that the heme and Cys72 constitute a redox couple [65]. Mutation of Cys72 blocks the production of Fe(III)-NO, further confirming the role of Cys. Based on absorption spectroscopy and stopped flow spectroscopy, it was suggested that the Fe(III)–NO complex formed through the NOD reaction (Reaction 4) followed by NO binding to the Fe(III) heme (Reaction 3) can produce heme-dependent S-nitrosylation of Cys72, with a concomitant reduction in the heme to the Fe(II) form (Reaction 10) followed by O_2_ binding [65]. The binding of O_2_ under almost anaerobic conditions and in the presence of relatively high concentrations of NO is only possible due to its unusually high affinity for *Ascaris* Hb. Finally, Cys-SNO can be reduced to Cys through a yet-unknown mechanism, thus allowing the enzyme cycle to proceed (Figure 5).

### 3.3. Myoglobin

Mb is a monomeric globin present in the cardiac and aerobic skeletal muscles of almost all vertebrates, where it plays a role in O_2_ storage and NO homeostasis [123]. The high O_2_ affinity of Mb maintains a robust gradient in pO_2_ from blood to myocytes, thus channeling O_2_ from the bloodstream to the mitochondria for oxidative phosphorylation. Some fish species, such as threespine stickleback (*Gasterosteus aculeatus*), have lost the ability to produce Mb [124] and 6 of the 16 species of icefishes lack the expression of Mb in the heart ventricle due to 4 independent mutations [125]. Mb is also largely absent from the heart of the African butterflyfish (*Pantodon buchholzi*), but is highly expressed in the swim bladder [124]. The discovery of Mb in smooth muscle, endothelial, and tumor cells [126,127,128] and the characterization of Mb-knockout mice [129] have suggested that Mb might play a more complex and versatile role than previously thought.

#### 3.3.1. Heme Reactivity with NO

The reactivity of mammalian Mbs with NO is one of the most studied systems in biochemistry. Under normoxic conditions, when the O_2_-bound form dominates, Mb scavenges NO producing Fe(III)-Mb and NO_3_^−^ through the NOD reaction (Reaction 4) [67]. Under hypoxic conditions, where Mb becomes increasingly deoxygenated, Mb produces NO from NO_2_^−^ through the NiR reaction (Reaction 6), faster than Hb [68]. Therefore, Mb has a dual role of NO scavenger and NO producer at different pO_2_ [130], thus regulating the activation of the sGC signaling pathway [131]. The NiR activity of Mb in the myocardium was shown to be the major source of NO [130] and has a relevant function under hypoxic conditions and in ischemia-reperfusion events [132]. Therefore, NO_2_^−^ can act as a stable reservoir for supporting NO signalling in hypoxic or metabolic stress.

#### 3.3.2. S-Nitrosylation

Human Mb has a single reactive Cys in position 110 [133] known to form Cys-SNO in vitro through metal-to-Cys S-nitrosylation, reacting with NO when the protein is in the Fe(III) form, similarly to Hb (Reaction 10) [67,134,135]. While S-nitrosylation in Hb was proposed by some authors to be dependent on the allosteric R–T transition, the degree of O_2_ saturation does not affect the accessibility of Cys110 for S-nitrosylation. Rayner et al. [135] demonstrated the S-nitrosylation of human Mb-Fe(II)-O_2_ in vivo, suggesting that the modified protein may stimulate aortic vessel relaxation similar to free NO. Mb-SNO may also be important in maintaining optimal NO concentrations in the human vasculature.

Teleosts Mbs, in which S-nitrosylation has been well documented [129,136,137], exhibit the same globin-fold as that of higher vertebrates but lack the D-helix [138]. Among teleosts, Antarctic notothenioids, carps, and tunas have a conserved Cys—Cys10—in the A helix, whereas the non-tuna scombroids, bonito, mackerel, and blue marlin lack this residue [137]. Cys10 is partially solvent-exposed and is located between the side chains of Asp118 and Lys9 at the protein surface, thus supporting the hypothesis that the “acid-base” motifs are important determinants for Cys S-nitrosylation [69]. The X-ray crystal structure of S-nitrosylated blackfin tuna Mb (PDB: 2NRM) (Figure 4b) indicates that S-nitrosylation at Cys10 causes reversible conformational changes [69]. Recently, the spectral, dynamical, and structural properties of unmodified blackfin tuna Mb and its S-nitrosylated derivative were studied by molecular dynamic simulations, suggesting that S-nitrosylation can also modulate local hydration. Indeed, the local hydration between the A and H helices and around the site of the S-nitrosylation is increased in S-nitrosylated Mb compared to the unmodified protein, with changes in the structure and dynamics of the protein [70].

Mbs of salmonids—among which rainbow trout and Atlantic salmon have been particularly investigated—were shown to undergo S-nitrosylation at a Cys residue different from Cys10 [71,72]. Indeed, in addition to Cys10, they exhibit a second Cys residue in position 107, located at the interface between the G and H helices, far from the heme pocket. The S-nitrosylation of these Mbs induces conformational changes in the tertiary structure associated with an increase in the O_2_ affinity [71,72]. Conversely, under hypoxic conditions, NO is released, improving myocardial efficiency during intense exercise [71,72].

### 3.4. Neuroglobin

Ngb is a globin predominantly expressed in neurons. Sequence conservation and its presence in several taxa point to an important role in the central nervous system, which, however, has not been fully elucidated [139]. Ngb has been associated with O_2_ supply to tissues, O_2_-scavenging, and sensing. It exhibits an hexacoordinated heme both in the Fe(II) and Fe(III) forms [140,141].

#### 3.4.1. Heme Reactivity with NO

Ngb in the Fe(II) form binds NO under anaerobic conditions (Reaction 2) upon displacement of the distal His. The complex has an estimated dissociation rate constant of 2.9 × 10^−3^ s^−1^ and a K_D_ of 7.7 × 10^6^ M^−1^, lower than that of Mb or Hb by a factor of 10^4^–10^5^. [74]. In the Fe(III) form, Ngb reacts with NO, leading to autoreduction (Reaction 5) followed by the binding of free NO to Fe(II) [73]. The NOD reactivity (Reaction 4) was also investigated, indicating that it occurs in a biphasic fashion consistent with the formation of a peroxynitrite intermediate [73]. It was also shown that Ngb efficiently catalyzes the conversion of NO_2_^−^ to NO with a second-order rate constant of 5.1 M^−1^ s^−1^ (Reaction 6, NiR reaction), although the role of this reaction in the retina and brain is not clear [74].

#### 3.4.2. S-Nitrosylation

The S-nitrosylation of Ngb has been observed for the murine ortholog upon reaction with NO_2_^−^ [74], hinting at heme-dependent S-nitrosylation (NiR reaction, Reaction 6, followed by metal-to-Cys S-nitrosylation, Reaction 10). Murine Ngb has two Cys residues, Cys55—only ∼15 Å from the heme iron—and the solvent-exposed Cys120. Although the exact site of S-nitrosylation has not been defined, Cys55 was deemed the most likely one [74].

It was hypothesized that NO^+^ generated by NO reaction with Fe(III) heme may be a possible pathway of S-nitrosothiols formation [74]. Alternatively, the reaction might occur through N_2_O_3_ (Reaction 9), which could be accommodated in a large hydrophobic cavity present in Ngb and connecting the heme distal site to the solvent [74]. The levels of S-nitrosylated Ngb formed in vitro under physiological concentrations of NO_2_^−^, however, were deemed too low to function as NO storage, but it was hypothesized that it could have a potential role in redox signalling pathways [74].

### 3.5. Cytoglobin

Cygb is a globin discovered 20 years ago [142] expressed at low concentrations in the cytoplasm of all tissues of vertebrates. It is upregulated under hypoxic conditions [143]. Like Ngb, Cygb is hexacoordinated, with two His residues coordinated at the proximal and distal sites [144]. Several functions have been proposed for Cygb, including NO scavenging [145], scavenging of ROS [146], lipid peroxidation [147], and synthesis of NO through the NiR activity [75].

#### 3.5.1. Heme Reactivity with NO

Similarly to other globins and heme proteins in general, Cygb in the Fe(II) form binds NO under anaerobic conditions (Reaction 2) [75] and catalyzes the NOD reaction under aerobic conditions (Reaction 4), with a comparable rate as Ngb [33]. The Leu46Trp variant decreases NOD activity > 10,000-fold. After conflicting reports [74], it was shown that under anaerobic conditions Cygb generates NO through the NiR activity (Reaction 6), with the formation of Cygb-Fe(III) and Cygb-Fe(II)-NO [75]. It was suggested that this reaction occurs in vivo and is crucial in the activation of sGC under hypoxic conditions [75]. Cygb NiR activity is strongly dependent on the oxidation state of the two surface-exposed Cys38 and Cys83, with the formation of the disulfide bond increasing the activity 50-fold [7]. Among teleosts, the heme of Cygb1 of Danio rerio can efficiently reduce NO_2_^−^ to form NO (Reaction 6, NiR reaction) at a rate of 14.2 M^−1^s^−1^, compared to 0.40 M^−1^s^−1^ of mammalian Cygb and 0.31 M^−1^s^−1^ of fish Cygb2 [76].

#### 3.5.2. S-Nitrosylation

Since the redox state of Cys38 and Cys83 has been associated with the NiR activity of Cygb, it appears plausible that the S-nitrosylation of one or both residues could contribute to modulating the activity. Indeed, it was shown that Cygb undergoes S-nitrosylation upon incubation with NO_2_^−^, albeit at lower levels in comparison with Ngb [74]. Both Cys38 and Cys83 are located in proximity to the internal cavity [148]. It was hypothesized that the formation of S-nitrosylated Cys may be heme-independent or that it could result from the reactivity with N_2_O_3_ [74]. As for Ngb, the levels of S-nitrosylated Cygb are likely too low to function as NO storage system [74].

### 3.6. NO Synthase

In metazoans, NO is produced by three isoforms of NO synthase (NOS) —the neuronal NOS (nNOS or NOS I), the inducible NOS (iNOS or NOS II), and the endothelial NOS (eNOS or NOS III). All contain a heme group. All NOS use L-arginine and O_2_ as substrates to generate L-citrulline and NO. NOS require NADPH, flavin adenine dinucleotide (FAD), flavin mononucleotide (FMN), and (6R)5,6,7,8-tetrahydrobiopterin (BH4) as cofactors [24,149]. NOS I is constitutively expressed in central and peripheral neurons, NOS II is expressed in many cells, and NOS III is mostly expressed in endothelial cells [150]. All vertebrate NOS are homodimers linked to a zinc ion by tetrahedral coordination. Each NOS monomer is endowed with an N-terminal oxygenase domain and a C-terminal reductase domain, both linked by a calmodulin (CaM) binding site. The Fe(II) heme binds O_2,_ which oxidizes the guanidine group of the L-arginine to generate L-citrulline and NO in a two-step reaction:

2 L-arginine + 3 NADPH + 3 H^+^ + 4 O_2_ → 2 citrulline +2 NO + 4 H_2_O + 3 NADP^+^

The NOS-derived NO binds the Fe(II)-heme group of sGC. The formation of a NO-ferroheme–sGC complex induces the synthesis of the second messenger cyclic guanosine 3′,5′-monophosphate (GMP).

#### 3.6.1. Heme Reactivity with NO

Similarly to other NO-hemoproteins, NOS may serve as a reservoir of bioactive NO [131]. Upon synthesis and before being released, the newly synthesized NO binds to the heme of NOS [151,152], forming the heme Fe(III)–NO complex (Reaction 3). This complex can release NO with a rate constant k_d_ (productive cycle) or it can be reduced by the NOS reductase domain to Fe(II)–NO through the rate constant k_r_, comparable to k_d_. Fe(II)–NO can be oxidized in the presence of O_2_ (futile cycle) through the k_ox_ rate constant [77,153].

The k_ox_ constant, much higher in NOS enzymes than in any other hemoprotein [77], is characteristic of heme-thiolate enzymes and does not require NO dissociation from the heme before the NOD reaction can take place (Reaction 4). Indeed, it occurs by the direct reaction of O_2_ with the heme Fe(II)–NO complex, through an N-bound Fe(III)-N(O)OO- intermediate species [78,154,155] (Figure 6).

#### 3.6.2. S-Nitrosylation

Constitutive NOS (nNOS and eNOS) are S-nitrosylation targets of NO themselves [155,156], with eNOS constitutively S-nitrosylated in endothelial cells. S-nitrosylation reversibly inhibits eNOS activity and is responsible for its fine regulation [157,158], supporting the hypothesis that S-nitrosylation is a dynamic and physiologically relevant regulator of NO signaling pathways in the vascular endothelium. The S-nitrosylation of Cys98 is involved in the disruption of the zinc tetrathiolate cluster of the eNOS dimer [81].

In further studies of human eNOS, multiple Cys residues, all located on the surface of the protein, were identified as potential sites of S-nitrosylation. Among these, Cys93 and Cys98 are involved in the formation of eNOS homodimers through a Zn tetrathiolate cluster [80]. Other Cys residues (Cys660, Cys801, and Cys1113) located within regions known to bind FMN, FAD, and NADPH, as well as Cys852, Cys975/990, and Cys1047/1049 located in regions not yet identified in any known biochemical functions can undergo S-nitrosylation [80].

### 3.7. Guanylyl Cyclase

Soluble guanylyl cyclase (sGC), a heme-containing heterodimer composed of an α and a β subunit, is the main receptor of NO in mammals. Each subunit contains three domains: an N-terminal portion consisting of a heme–NO/O_2_ (HNOX) region with the heme present only in the β subunit; a “dimerization domain” with a Per/Arnt/Sim (PAS)-fold domain and a coiled-coil region (CC domain); and the C-terminal catalytic domain whose activation occurs when the α and β subunit are associated [159]. The HNOX and PAS domains form the N-terminal sensor module of sGC whereas the CC domains of both subunits form the transducer module that connects the N-terminal sensor module and the C-terminal catalytic module. NO has a dual effect on sGC: it binds at the heme and stimulates its catalytic activity and reacts with a Cys residue producing S-nitrosylation, which inhibits the enzyme.

#### 3.7.1. Heme Reactivity with NO

When NO binds the heme of the β subunit, the domain of sGC catalyzes the cyclization of guanosine triphosphate (GTP) to the second messenger cyclic cGMP, initiating the signal transduction cascade [160,161]. In the HNOX domain of the β-subunit, the b-type heme is in the pentacoordinated Fe(II) state and exhibits a high affinity for NO (Reaction 2) [162]. The proximal ligand of the heme is His105, whose binding with Fe(II) is broken when NO binds the iron on the distal side of the heme [82,83]. Upon the binding of NO to the β heme of human HNOX, structural changes in the sensor module trigger a conformational switch of the transducer module from bending to straightening, inducing structural changes within the catalytic module and activating the enzyme [84].

#### 3.7.2. S-Nitrosylation 

In addition to the binding of heme, sGC can be S-nitrosylated, as observed in smooth muscle cells [86]. S-nitrosylation of sGC affects its sensitivity to NO at the heme group, resulting in decreased responsiveness. In particular, Cys122 in the β subunit and Cys243 in the α subunit were identified as potential targets for S-nitrosylation. Desensitization of sGC deriving from S-nitrosylation constitutes a mechanism of sGC modulation by negative feedback [86]. sGC can also undergo a reductive nitrosylation reaction (Reaction 5) coupled with the heme-dependent S-nitrosation of Cys residues [85].

Oxidation of the heme iron in sGC causes dysfunction in NO/cGMP signal transduction, with effects on cardiovascular physiology as it decreases the affinity for NO and the enzymatic response to NO [163]. Fe(III) sGC undergoes reductive nitrosylation with NO in two steps, requiring two equivalents of NO. The first NO reduces the iron from Fe(III) to Fe(II) (Reaction 5, autoreduction in Fe(III)-NO). The second equivalent of NO binds the heme iron, forming a Fe(II)–nitrosyl complex (Reaction 2, binding of NO to Fe(II)). Reductive nitrosylation is accompanied by a conformational change in the protein and a heme-dependent SNO formation of Cysβ78 and Cysβ122 with the NO desensitization of Fe(III) sGC [85]. The Trx1/TrxR system may be involved in restoring the basal activity of sGC and its NO sensitivity via a mixed disulfide exchange mechanism between sGC and Trx1 [164].

The modulation of sGC activity by thiol oxidation has been proposed as a therapeutic approach for pathologies associated with oxidative or nitrosative stress, such as cardiovascular diseases [159].

### 3.8. Cyclooxygenases

Cyclooxygenases (COX), previously known as prostaglandin H synthases, are homodimeric enzymes that catalyze the first, rate-limiting step of prostanoids biosynthesis from arachidonic acid through their dual peroxidase and dioxygenase activities [165]. COXs consist of three domains: the epidermal growth factor domain, the membrane-bound domain, and the heme-binding catalytic domain, which contains the cyclooxygenase and peroxygenase active sites [165]. The heme group is in the Fe(III) in the resting oxidation states of COXs. Their reaction is multistep. First, COXs oxygenate arachidonic acid to prostaglandin G2, which is then reduced to prostaglandin H2 through the peroxidase activity in the enzyme.

Two homologous COXs have been identified, COX1 and COX2. COX1 is expressed constitutively, whereas the expression of COX2 is upregulated by inflammatory and proliferative stimuli. The interaction between NO and COXs and the activation of recombinant COX2 by NO have long been known [166], although the dissection of the effects of S-nitrosylation and NO heme binding has been carried out more recently, pointing to the former as the main effect of NO on the protein function [167].

#### 3.8.1. Heme Reactivity with NO

Early spectroscopic experiments suggested that NO does not bind the Fe(III) heme group of COX1 [168]. NO binding to the heme of COX1 was monitored by stopped-flow spectrophotometry and enzyme assays. A very high dissociation constant of 0.92 mM was estimated [87]. However, the reaction was much faster with the Fe(II) form [87]. Even upon binding, NO only produced a slight decrease in enzyme activity, ruling out the regulation by direct nitrosylation at the heme [87]. The limited binding of NO to the heme group was also observed for COX2 in the Fe(III) state [167].

#### 3.8.2. S-Nitrosylation

COX undergoes several post-translational modifications [169]. Particularly, S-nitrosylation was observed in both COX1 and COX2, although only the latter undergoes activation as a consequence [167]. NO was found to increase COX2 activity by increasing the apparent V_max_ 1.6-fold and by decreasing the K_M_^O2^ 1.5-fold, only in the presence of O_2_ [167]. Consistently, the conformation of COX2—but not COX1—was shown to be altered by S-nitrosylation in circular dichroism experiments [167]. The Snyder group showed that COX2, when incubated with various nitroso donors, is S-nitrosylated and activated in HEK293T cells overexpressing COX2 [88]. By individually substituting the 13 Cys residues of COX2 to Ser, they concluded that Cys526 is the likely site of allosteric activation, although other Cys residues were also S-nitrosylated [88].

By investigating the truncated variants of COX2 and iNOS in a murine macrophage cell line, it was concluded that the two proteins directly interact, bringing the NO-producing enzyme in proximity to COX2 for S-nitrosylation [88]. It was also shown that COX2 also binds nNOS and that the disruption of the complex prevents N-methyl-D-aspartate (NMDA) neurotoxicity [170]. In other experiments, COX2 was shown to coimmunoprecipitate with iNOS but not with eNOS [171]. In rats, atorvastatin, a statin with cardioprotective activity, causes the S-nitrosylation of COX2 through iNOS [171]. Atorvastatin-induced S-nitrosylation of COX2 contributes to the production of omega-3 fatty acids-derived protective molecules known as 13-series resolvins [172]. Overall, the interaction of NOS and COX2 constitutes a layer of the complex interplay between prostaglandin biosynthesis and NO signaling [173].

No effect has ever been observed for COX1 in vitro when incubated with either NO or nitroso donors [87,167]. However, it was shown that bombesin might induce the S-nitrosylation and activation of COX1 in presympathetic spinally projecting neurons in a rat model, thus eliciting the secretion of catecholamines [174,175]. It was also shown that COX1, unlike COX2, does not coimmunoprecipitate with iNOS [176].

### 3.9. Nitrophorins (Cimex lectularius)

NPs are heme proteins discovered 30 years ago [177] in the saliva of blood-feeding insects. They include proteins form unrelated lineages in a typical case of convergent evolution. *Rhodnius prolixus* NP exhibits a lipocalin-like eight-stranded β-barrel structure, whereas *Cimex lectularius* NP has a β-sandwich motif with seven peripheral α-helices [6,178]. The heme iron of NPs is stable in the Fe(III) state and binds NO (Reaction 3) in a pH-dependent fashion [31]. Upon contact with blood, which exhibits a higher pH than the insect’s saliva, the NO–NP bond weakens, and NO is released in the capillaries, thus facilitating the insect’s feeding through vasodilation, reduced platelet aggregation, and reduced blood coagulation [179]. The heme of *C. lectularius* NP—the NP for which S-nitrosylation has been observed in addition to NO heme binding—is located between the β-sandwich and one of the α-helices, and it is coordinated to Cys60, forming a Fe(III)-thiolate complex. At the distal side, the heme is partially coordinated with a water molecule as a sixth ligand. Consistently, the crystal structure of *C. lectularius* NP shows a weakly bound water molecule on the distal side [90].

#### 3.9.1. Heme Reactivity with NO

Since NO transport and delivery is the main physiological role of *C. lectularius* NP, the properties of the complex have been thoroughly investigated [31,89]. *C. lectularius* NP forms a stable heme Fe(III)–NO complex—with a K_d_ in the µM to nM range—and NO can, therefore, be stored for a long period of time in the salivary gland of the insect.

#### 3.9.2. S-Nitrosylation

At high NO concentrations, it was observed that *C. lectularius* NP is S-nitrosylated both at the heme and at the proximal Cys, Cys60 [90]. At NO concentrations in the 200 μM–2 mM range, a first molecule of NO binds the Fe(III) heme, and, following the homolytic cleavage of the Fe(III)−S bond, a second molecule reacts with Cys60 (Figure 7). *C. lectularius* NP can, therefore, reversibly bind two molecules of NO at two different sites. At lower NO concentrations, NO is first released from the Cys60-SNO adduct following the transfer of an electron from Fe(II). In parallel, Fe(II) is oxidized to Fe(III), which exhibits a lower affinity for NO—particularly at a higher pH—thus releasing the second molecule of NO [90]. The S-nitrosylation of Cys60 was also confirmed by X-ray crystallography (PDB: 1Y21) (Figure 4c) [180]. This reactivity is not shared by another Cys-coordinated heme protein, cytochrome P450, in which the proximal cysteinate ligand is protected by a network of hydrogen bonds [181].

The mechanisms proposed for Cys60 S-nitrosylation are essentially two [31]: **(i)** a concerted mechanism in which NO attacks the cysteinate concomitantly with the cleavage of the Fe(III)–S-Cys60 bond, or **(ii)** a two-step mechanism in which the homolytic cleavage transiently forms the thiyl radical (RS•), which then reacts with a second molecule of NO to form SNO. The latter mechanism is deemed less likely [31]. It is currently under debate if the release of NO by Cys60 is physiologically relevant, since it would require high NO concentrations. Moreover, the Fe(II)–NO heme, whose formation should be coupled with S-nitrosylation (Figure 7), has not been detected in electron paramagnetic resonance (EPR) experiments on whole-gland homogenates at pH 7.0 [180].

### 3.10. Plant Ascorbate Peroxidase

Ascorbate peroxidase (or L-ascorbate peroxidase, APX) (EC 1.11.1.11) converts ascorbate to dehydroascorbate using a molecule of hydrogen peroxide as an electron donor. It is a central component of the ascorbate–glutathione cycle, and its activity was first reported in 1979 [182]. It is a member of the family of heme-containing peroxidases, which catalyze the H_2_O_2_-dependent oxidation of a wide range of substrates. APX is an important component of the enzymatic antioxidant defense system in plants in several organelles, including chloroplasts, cytosol, mitochondria, and peroxisomes [183].

#### 3.10.1. Heme Reactivity with NO

NO is known to bind tobacco APX at the Fe(III) heme group (Reaction 3), inhibiting its activity in a reversible fashion [91]. The reaction of tobacco APX with a short-lived NO donor (NOC-9) demonstrated a maximum inhibition of 66% of APX after a 5-min incubation, whereas after 60 min from its removal fully restored its activity, providing evidence that NO inhibits APX in a reversible manner [91].

#### 3.10.2. S-Nitrosylation

In plants, NO is involved in the regulation of various important processes, including hormone signaling, stress responses, and cell death [184]. In plants, many proteins are modified by S-nitrosylation, which directly regulates downstream the activity in the target proteins [184]. However, the mechanisms of NO production in photosynthetic organisms are still controversial since no NOS has been identified [185]. Therefore, the NO_2_^−^-dependent pathway is probably the main source of NO production in plants [186].

The peroxidase activity in cytosolic APX is regulated by post-translational modifications such as S-nitrosylation, tyrosine (Tyr) nitration, and S-sulfhydration [187]. Proteomics data show that cytosolic APX is the main potential target of S-nitrosylation. The Arabidopsis genome contains nine genes encoding APX, of which APX1, APX2, and APX6 are cytosolic enzymes, and the other six enzymes are found in other subcellular compartments [188]. APX1 is involved in hydrogen peroxide detoxification using ascorbate as reducing equivalents. NO coordinates the intracellular level of ROS during stress responses through the S-nitrosylation-mediated boosting of the H_2_O_2_-scavenging activity of APX1 and attenuation of the ROS synthesis activity of NADPH oxidase [92]. Arabidopsis APX1 contains five Cys residues, two of which (Cys32 and Cys49) were identified as S-nitrosylated by biochemical and homology modeling studies. Homology modeling predicts that Cys32 and Cys49 are accessible, thus facilitating S-nitrosylation. The S-nitrosylation of APX1 at Cys32 was shown to enhance the enzymatic activity of the protein, leading to the increased resistance to oxidative stress, whereas the substitution of Cys32 with Ser causes a nearly 50% reduction in the S-nitrosylation of the APX1 mutant and, consequently, the reduction in ascorbate peroxidase activity [92]. Moreover, the S-nitrosylation of APX1 at Cys32 also plays an important role in regulating immune responses [93]. Recent evidence suggests that the S-nitrosylation of APX in *Nicotiana tabacum* may induce APX ubiquitination and degradation [93], an opposite behaviour compared to more recent data in Arabidopsis, which show that the S-nitrosylation of Cys32 causes an increase in APX activity, thus suggesting an interplay between NO and ROS metabolism.

### 3.11. Catalase

CATs catalyze the dismutation of H_2_O_2_ into H_2_O and O_2_. CATs are found in diverse organisms, including prokaryotes, fungi, animals, and plants, and are tetramers. CATs are highly expressed and constitute an integral part of the plant antioxidative system [189]. Mammalian CAT is a homotetrameric peroxisomal heme-containing enzyme essential for scavenging oxidative species. Each monomer contains a heme group with the iron in the Fe(III) state. All CATs contain the heme group essential for enzymatic activity with an absorption maximum around 404–406 nm [190]. As an APX, CATs are characterized by their high specificity for H_2_O_2_, and weak activity against organic peroxides. The second type of heme-dependent CAT is the bifunctional CAT-peroxidases found in some fungi and cyanobacteria [191]. Bifunctional CAT-peroxidases are more similar to APX and fungal cytochrome c peroxidase. Available genomic information suggests that most animals, including mammals, contain a single CAT gene, whereas the Arabidopsis genome is endowed with three genes (CAT1, CAT2, CAT3) [192]. CATs may also be grouped into classes. According to this grouping, Class I CATs are strongly expressed in photosynthetic tissues, while Class II CATs are associated with vascular tissues. Class III CATs are notably expressed in seeds and reproductive tissues.

#### 3.11.1. Heme Reactivity with NO

CAT was the first antioxidant enzyme found to be modulated by NO donors and it was demonstrated that NO inhibits the activity in tobacco CAT, which is also irreversibly inhibited by peroxynitrite [91]. As in mammals [193,194], NO has been shown to reversibly inhibit CAT activity by directly interacting with the heme prostetic group. The reaction of tobacco CAT with NO donor (S-nitroso-N-acetylpenicillamine, SNAP) shows an initial absorption spectrum with a peak of around 405 nm, consistent with the enzyme in the Fe(III) state; the final spectrum, after incubation with SNAP for 3 h, shows an increased absorbance peak at 420–440 nm, due to the NO binding to the heme, accompanied by the inhibition of enzyme activity. After removing SNAP, there is a recovery of active CAT FeIII with a peak of 405 nm, demonstrating that the inhibition is reversible [91]. More recently, the X-ray structure of bovine liver CAT complexed with NO (CAT-NO) showed that, like Fe(III)-Mb-NO, CAT-NO is comparatively labile and that the heme iron in CAT may be endowed with some metastable Fe(II) form, despite being formally Fe(III) [195]. 

#### 3.11.2. S-Nitrosylation

There is emerging evidence that plant CATs are S-nitrosylated in vitro [91,196], although the specific target residues have not yet been identified [197]. In Ganoderma lucidum, a well-known medicinal mushroom, the S-nitrosylation of CAT was potentially identified at three sites—Cys401, Cys642 and Cys653—by mass spectrometry [94]. The authors also showed that CAT is S-nitrosylated (in vitro and indirectly in vivo by mutants of GSNOR). These S-nitrosylated Cys residues may interfere with the CAT activity regulated by GSNOR. In Arabidopsis, GSNOR (AtGSNOR1) is the major regulator of the total cellular SNO levels in plants and the master regulator for intracellular reactive nitrogen species (RNS) levels. The repressor of GSNOR1 (ROG1), identified as the non-canonical CAT (CAT3), exhibits transnitrosylase activity that specifically modifies AtGSNOR1 at Cys10. ROG1 and GSNOR1 cooperate to modulate the intracellular RNS levels, thus regulating NO-based redox signaling in plants [198].

### 3.12. Cytochrome c

Cytochrome c is a highly conserved electron carrier, present in both prokaryotic and eukaryotic organisms. It is fine-regulated by many post-translational modifications that can alter its structure and function, leading to a variety of conformational states of biological relevance [199,200]. Cytochrome c is a small heme protein of 13 kDa, localized between the inner and outer mitochondrial membranes, where it is involved as an electron transfer mediator of the respiratory chain for the synthesis of ATP [201]. In its heme c, the central iron atom is hexacoordinated with four nitrogens from the protoporphyrin IX and two axial ligands: the His18 and the sulfur atom of a methionine (Met80) side chain [202,203]. These axial ligands provide the appropriate redox potential for electron transfer from the electron donor of complex III (cytochrome bc1 complex or CoQH2-cytochrome c reductase) to the electron acceptor of complex IV (cytochrome c oxidase) to cytochrome c oxidase in the respiratory chain [199].

Cytochrome c is also involved in apoptotic pathways. In fact, after an apoptotic stimulus, it is released from mitochondria into the cytoplasm, stimulating apoptosome formation and subsequent caspase-3 activation and, thus, leading to the propagation of the apoptotic cascade [201,204,205].

#### 3.12.1. Heme Reactivity with NO

During apoptosis, Fe(III) cytochrome c can be heme-nitrosylated (Reaction 3) and then released into the cytoplasm. This heme nitrosylation increases caspase-3 activation whereas the inhibition of intracellular cytochrome c nitrosylation decreases the cascade of apoptosis [95]. Having a hexacoordinated heme less reactive with NO, during apoptosis, cytochrome c may be subjected to a conformational change resulting in a pentacoordinate heme [206]. In the Fe(II) form, the reaction of NO with cytochrome c induces the cleavage of an Fe–Met80 bond, allowing the binding of the Fe(II) heme to NO (Reaction 2) [96].

#### 3.12.2. S-Nitrosylation

Since cytochrome c has no free Cys, it can be nitrosylated only on its heme. However, cytochrome c is a mediator of S-nitrosation in biological systems, under anaerobic conditions [97] and aerobic conditions in living cells [98]. In fact, Fe(III) cytochrome c can act as an electron acceptor allowing the formation of GSNO and Fe(II) cytochrome c, starting from GSH and NO. GSNO can participate in transnitrosation reactions and transfer a reversibly S-nitroso functional group to another thiol [207,208]. Lower SNO formation has been found in cytochrome c deficient mouse embryonic cells as compared to wild-type controls.

### 3.13. Cytochrome c Oxidase

Cytochrome c oxidase (CcOX), or Complex IV, is a transmembrane protein complex found in bacteria, archaea, and in the mitochondria of eukaryotes, where it is the terminal complex of the oxidative phosphorylation pathway. It receives electrons from four cytochrome c molecules and transfers them to one O_2_ molecule, producing two molecules of water. Concomitantly, it transports four protons across the membrane, contributing to the proton gradient used by ATP synthase to produce ATP. The subunit composition of the mitochondrial enzyme depends on the species and can include between 5 and 13 subunits. The three major subunits (I, II, III) are encoded by mitochondrial DNA, whereas the other smaller subunits are encoded by the nuclear genome and synthesized on cytoplasmic ribosomes. Subunits I and II are the catalytic core of the enzyme. Subunit I contains heme α, heme α3, and CuB, while subunit II contains CuA and the cytochrome c binding site. Subunit III and most of the nuclear subunits are essential for the assembly of a functional catalytic enzyme. Heme α is coordinated by two His residues, whereas heme α3 by one His residue [209]. 

#### 3.13.1. Heme Reactivity with NO

NO is produced continuously in the mitochondrial membrane by mitochondrial NOS and it can inhibit the electron transport chain at multiple sites, the most sensitive of which is CcOX, with 50% reversible inhibition at NO concentrations of 0.1 µM [210]. According to Pearce et al. and Collman et al. [99,100], NO reacts with Fe(II) heme α3 to form an iron-nitrosyl complex with a high affinity (K_D_ = 0.1 nM) [211]. NO released from the heme binds rapidly to the cupric (CuB^2+^) site to reduce it to the cuprous state (CuB^+^) and produces nitrite (NO_2_^−^) [212]. Long-term exposure to pathophysiological concentrations of NO (>1 µM) leads to the persistent inhibition of cell respiration [213]. The physiological consequences of the interaction between CcOX and NO are dependent on the intracellular O_2_ concentration and the redox state of CcOX [214]. Inhibition clearly involves the competition between O_2_ and NO in binding to the bimetallic heme α3-CuB active site of the enzyme.

#### 3.13.2. S-Nitrosylation

Due to the high amount of nitrosylating agents, mitochondrial proteins are subject to S-nitrosylation, which mostly inhibits their activities. Mitochondrial proteins targeted for S-nitrosylation include all the complexes of the electron transport chain (ETC): Complex I (NADH: ubiquinone oxidoreductase); Complex II (succinate dehydrogenase); Complex III (cytochrome b-c1 Complex); cytochrome c, Complex IV (CcOX), as well as adenosine triphosphate (ATP) synthase (Complex V) [101]. Zhang et al. demonstrated that the long-term exposure of pulmonary artery endothelial cells to high concentrations of NO causes the modification of Cys residues located in subunit II of CcOX, revealing a novel putative NO-sensitive motif containing Cys196 and Cys200 [102]. Site-directed mutagenesis of these two Cys reduced NO-mediated nitrosylation of complex IV [102].

## 4. Conclusions

The interaction of NO with biological systems is pervasive throughout the evolutionary history of life and the understanding of its role in physiology and pathology can lead to the development of therapies for human diseases. Since the discovery of NO as EDRF, there has been an intense interest in how NO is produced, distributed, scavenged, and how it interacts with target proteins. Heme proteins were found to play a central role in the regulation of NO metabolism since the heme group is capable of both reversibly binding NO and catalyzing complex redox reactions that can result in both its scavenging and its synthesis. The axial ligands and the heme environment can drastically change these reactivities and, therefore, the physiological role of the specific heme protein. As several Cys-containing proteins, heme proteins are capable of forming Cys-SNOs, which can lead to transient changes in protein function. Heme proteins are unique in exhibiting this dual reactivity toward NO, with increasing evidence of cross-reactivity between the heme and Cys residues within the same protein, particularly through a reaction known as “heme-dependent S-nitrosylation”. This cross-reactivity can finely regulate the reactivity of heme proteins toward NO and other nitrogen compounds. However, since most heme proteins exhibit various reactivities with NO and that several Cys residues can be S-nitrosylated in vitro, it is imperative to assess the extent of these reactions in vivo before concluding that they are physiological relevant. The current debate on the interaction of Hb with NO is a clear example of this complexity. We hope this review will contribute to highlight the importance of NO chemistry in heme proteins.

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
