# Peer review of "NO and Heme Proteins: Cross-Talk between Heme and Cysteine Residues"

_antioxidants, 2023, doi:10.3390/antiox12020321_

Round 1
Reviewer 1 Report
The manuscript "NO and heme proteins: cross-talk between heme and cysteine residues" by Cinzia Verde, Daniela Giordano and Stefano Bruno performs a review of the literature regarding NO and heme proteins interplay, with a special emphasis on heme-dependent nitrosilation. This post-translational modification has been identified as an additional mechanism that modulates biological funtion and the authors perform an extensive and clear literature review on the topic.
This Reviewer rather enjoyed reading the manuscript and congratulates the authors for their work. The manuscript is very well written and organised, easy to follow and read, and has the potential to become a fundamental work for those studying heme proteins, heme coordination, NO regulation and nitrosylation.
This Reviewer was tempted to accept the manuscript "as is" for publication. However, has just a few minor suggestions for the authors:
1. Page 1, lines 29 to 34: while most people are acquainted with heme b and c, heme a and o are not so common/known; this Reviewer suggests the authors to add an additional Figure depicting the different heme moieties avoiding the reader to "leave" the manuscript searching for the chemical structure of these moieties;
2. Page 2, lines 43-59: for sake of completeness, a couple of sentences could be added mentioning the "gasotransmitters" CO and H2S as well as their interplay/paring up with NO;
3. Figure 2: the red background in the reactions circles is way too strong; moreover, the lettering looses itself in the middle of that strong colour; suggestion: tone down the red and add bold to the lettering;
4. Page 5, lines 191-198: the nitrite reduction is bacterial sources is completely omitted and, for sake of completeness, should be (even so briefly) mentioned (e.g., Pubmed IDs: 17139260 and 26362109);
5. Besides the enzymes mentioned in the manuscript, extensive work has also been done by Vicente and Giufré regarding s-nitrosylation in human cysthathione beta-synthase (e.g., Pubmed ID's 27039165 and 34573023); the authors could consider adding a sub-section tho the manuscript addressing this very important human enzyme;
MINOR ISSUES
1. Page 1, lines 29-32; page 2, line 47, and so throughout the text - heme a, b, c and o, etc, should all be in italic: heme a, b, c and o;
2. Page 1, lines 29-32 - heme a, b and o should be in small caps;
3. Table 1 and Figure 3 (caption): "pdb" should be spelled "PDB";
4. Page 10, line 362: please correct "th enitroo group";
5. Page 18, section 3.9 (title): please italicize the species name;
6. Page 20, line 797: please write in full extent the name of the enzymes and only then introduce the acronyms;
7. Page 23, line 908: please italicize "et al".
This Reviewer wishes the authors all the best for their future endeavours and a wonderful 2023.
Author Response
We would like to thank the Reviewer for his\her comments. We believed he\she helped improve our work considerably. We have addressed all comments in sequential order. Our responses are provided in italics and highlighted in yellow in the revised version of the MS. We have also corrected some mistakes along the text and re-formatted all references.
Reviewer 1
The manuscript "NO and heme proteins: cross-talk between heme and cysteine residues" by Cinzia Verde, Daniela Giordano and Stefano Bruno performs a review of the literature regarding NO and heme proteins interplay, with a special emphasis on heme-dependent nitrosilation. This post-translational modification has been identified as an additional mechanism that modulates biological funtion and the authors perform an extensive and clear literature review on the topic.
This Reviewer rather enjoyed reading the manuscript and congratulates the authors for their work. The manuscript is very well written and organised, easy to follow and read, and has the potential to become a fundamental work for those studying heme proteins, heme coordination, NO regulation and nitrosylation.
This Reviewer was tempted to accept the manuscript "as is" for publication. However, has just a few minor suggestions for the authors:
- Page 1, lines 29 to 34: while most people are acquainted with heme b and c, heme a and o are not so common/known; this Reviewer suggests the authors to add an additional Figure depicting the different heme moieties avoiding the reader to "leave" the manuscript searching for the chemical structure of these moieties;
We thank the Reviewer for pointing this out. In the main text, we have added a new Figure 1, showing the different heme moieties between heme b and c, heme a and o.
- Page 2, lines 43-59: for sake of completeness, a couple of sentences could be added mentioning the "gasotransmitters" CO and H2S as well as their interplay/paring up with NO;
We have added a brief observation in this respect (now at the bottom of page 2)
- Figure 2: the red background in the reactions circles is way too strong; moreover, the lettering looses itself in the middle of that strong colour; suggestion: tone down the red and add bold to the lettering;
We have corrected Figure 2 (Figure 3 in the revised version) as suggested by Reviewer 1.
- Page 5, lines 191-198: the nitrite reduction is bacterial sources is completely omitted and, for sake of completeness, should be (even so briefly) mentioned (e.g., Pubmed IDs: 17139260 and 26362109);
We have added a sentence and a reference in this respect
- Besides the enzymes mentioned in the manuscript, extensive work has also been done by Vicente and Giufré regarding s-nitrosylation in human cysthathione beta-synthase (e.g., Pubmed ID's 27039165 and 34573023); the authors could consider adding a sub-section tho the manuscript addressing this very important human enzyme;
We could not find an indication that cystathionine beta synthase is S-nitrosylated. The lyase, oin which there is a work on S-nitrosylation, is not a heme protein and was therefore excluded from this review.
MINOR ISSUES
- Page 1, lines 29-32; page 2, line 47, and so throughout the text - heme a, b, c and o, etc, should all be in italic: heme a, b, c and o;
Done
- Page 1, lines 29-32 - heme a, b and o should be in small caps;
Done
- Table 1 and Figure 3 (caption): "pdb" should be spelled "PDB";
Done
- Page 10, line 362: please correct "th enitroo group";
Done
- Page 18, section 3.9 (title): please italicize the species name;
Done
- Page 20, line 797: please write in full extent the name of the enzymes and only then introduce the acronyms;
Done
- Page 23, line 908: please italicize "et al".
Done
This Reviewer wishes the authors all the best for their future endeavours and a wonderful 2023.
We thank the Reviewer for the time spent in reading and reviewing the manuscript. We are glad that the review has been appreciated by her/him.
Reviewer 2 Report
Heme proteins play major roles in NO signaling. Both NO synthesis as well the main target for NO are through heme proteins: NOS and sGC. Moreover, there is substantial data showing that thiol modifications via S-nitrosation or otherwise influence NO signaling. Thus, this review addresses an important topic. The review is interesting and covers a lot of material but could be improved by addressing the following:
(1) Some additional/alternative context might be given for Reaction 1 on line 77 of page 2. The authors write that NO’s half life is limited by reaction with oxygen. However, they should include the rate constant governing Eq 1. Wink and Ford found this to be about 10^6 (1/M^2)(1/s). At physiological NO, this reaction is quite slow! Other reactions would seem to define NO’s half -life. Moreover, how nitrite is made from NO in vivo is a bit of a mystery although reaction with ceruloplasmin has been suggested. Perhaps other copper proteins are involved as mentioned by the authors on page 22 for Cyt C oxidase.
(2) In discussing the reactions shown on page 4, it would be useful to have some rate constants.
(3) The dioxygenation reaction is mistakenly referred to as deoxygenation. This occurs on page 4 on lines 155 and 168 and possibly elsewhere. NO is dioxygenated (two oxygens are added) to form nitrate. Deoxygenation refers to when a oxygen dissociates from the heme. I think John Olson who studied this reaction extensively first used the term dioxygenation. See for example Free Radical Biology & Medicine, Vol. 36, No. 6, pp. 685 – 697, 2004
(4) More clarity may be added when discussing reaction 5 on page 5. Excess NO is not needed for the reduction step, but only for the nitrosylation step which is not shown.
(5) On page 9, the final sentence might need to be altered. The authors suggest that significant amounts of Hb(Fe(II)-NO) may not form in vivo due to competition with oxygen. However, in vivo, during much of the circulation, there are inbound hemes and these can and do bind NO. It would good to give numbers for affinities in general NO binds the ferrous heme about 10^5 times as strongly – so 1 nM NO competes with 100 uM oxygen. NO has ben detected on the heme by chemiluminescence and on EPR after NO inhalation. NO-heme is probably limited by the dioxygneation reaction in vivo.
(6) Although the inclusion of work on the B93 mutant mouse is a step in the right direction, a more fair discussion on the SNO-Hb hypothesis is needed. Work by the Gladwin group showing no gradient in SNO-Hb should be mentioned (PNAS, 10;97(21):11482-7), contrary to the statement that the amount of SNo changes made by the authors on line 381 of page 10. In addition, two papers showing no heme to Cys transfer upon allosteric changes should be cited and discussed: (PNAS, vol 100, 11303-11308, 2003 and Blood 2006;107:2602-2604). These papers show that when T is switched to R, or vice versa, the transfer does not occur. In scheme 4, the transfer is shown for a ferric heme nitrosyl. But the SNO-Hb hypothesis suggests this transfer occurs for the ferrous nitrosyl. The scheme 4 is chemically fine, but how much ferric heme nitrosyl is there in vivo! Greater than 2% metHb is pathological. Moreover, ferrous heme binds No way tighter than ferric heme. So if the transfer is supposed to occur from ferrous heme (as Stamler and colleagues propose), then where does the extra electron come from?
(7) Related to item 6, the authors discuss pathology of abhorrent transnitrosation starting in line 392 of page 11. If abherrant transnitrosation is so pathological, why doe the mutant mice (that have zero B93 SNO-Hb) have such a mild pathology?
(8) On page 12 when talking about the Ascaris Hb and reaction 4, the authors again refer to deoxygenation instead of dioxygneation and they also say this form nitrite instead of nitrate. Scheme 5 also incorrectly shows nitrite. This mistake is repeated on page 13 line 465. Dioxygenation makes nitrate not nitrite.
(9) Generally, reactions are referred to throughout much of the text but the reader is not likely to memorize which is which and therefore has to keep going back. It would be best to use the reaction numbers and names – ie nitrite reduction, ferrous heme NO binding, dioxygneation etc
(10) On page 14, line 529, the authors refer to NO+ generated from NO reaction with Fe(II) heme making SNO. This does not make sense and is not what Peterson et al discussed. It should be NO+ generated from nitrite reaction with Fe(II) heme
(11)In scheme 7, page 19, should the iron in the middle be FeIII instead of FeII?
(12) Some summary and conclusions would be a good addition rater than just ending in Cyt C oxidase
Typos
(1) Typo on line 362 of page 10
(2) Typo on line 478 page 13
Author Response
We would like to thank the Reviewer for his\her comments. We believed he\she helped improve our work considerably. We have addressed all comments in sequential order. Our responses are provided in italics and highlighted in yellow in the revised version of the MS. We have also corrected some mistakes along the text and re-formatted all references.